# Histaminergic System Activity in the Central Nervous System: The Role in Neurodevelopmental and Neurodegenerative Disorders

**DOI:** 10.3390/ijms25189859

**Published:** 2024-09-12

**Authors:** Dariusz Szukiewicz

**Affiliations:** Department of Biophysics, Physiology & Pathophysiology, Faculty of Health Sciences, Medical University of Warsaw, 02-004 Warsaw, Poland; dariusz.szukiewicz@wum.edu.pl

**Keywords:** histamine, histamine receptors, neurodevelopmental disorders, neurodegenerative diseases, histaminergic neurons, tuberomammillary nucleus, neuroinflammation, histamine H3 receptor, histamine H3 receptor antagonist/inverse agonist, autism spectrum disorders, Alzheimer’s disease

## Abstract

Histamine (HA), a biogenic monoamine, exerts its pleiotropic effects through four H1R–H4R histamine receptors, which are also expressed in brain tissue. Together with the projections of HA-producing neurons located within the tuberomammillary nucleus (TMN), which innervate most areas of the brain, they constitute the histaminergic system. Thus, while remaining a mediator of the inflammatory reaction and immune system function, HA also acts as a neurotransmitter and a modulator of other neurotransmitter systems in the central nervous system (CNS). Although the detailed causes are still not fully understood, neuroinflammation seems to play a crucial role in the etiopathogenesis of both neurodevelopmental and neurodegenerative (neuropsychiatric) diseases, such as autism spectrum disorders (ASDs), attention-deficit/hyperactivity disorder (ADHD), Alzheimer’s disease (AD) and Parkinson’s disease (PD). Given the increasing prevalence/diagnosis of these disorders and their socioeconomic impact, the need to develop effective forms of therapy has focused researchers’ attention on the brain’s histaminergic activity and other related signaling pathways. This review presents the current state of knowledge concerning the involvement of HA and the histaminergic system within the CNS in the development of neurodevelopmental and neurodegenerative disorders. To this end, the roles of HA in neurotransmission, neuroinflammation, and neurodevelopment are also discussed.

## 1. Introduction

Histamine (HA) is a nitrogenous compound (an imidazole ring attached to an ethylamine chain) with versatile biological effects, both local and systemic. Since its discovery in 1910 by Henry Dale and colleagues, HA has been considered a locally acting autacoid hormone closely associated with mast cells (MCs) as an inflammatory mediator released from their secretory granules in the course of the allergic reaction [1,2]. HA, as a participant in the immune response to foreign pathogens, is produced by basophils and MCs located in nearby connective tissue. A well-known effect of HA is an increase in the permeability of capillaries, allowing white blood cells and some proteins (e.g., glycoproteins such as immunoglobulins) to penetrate toward the pathogen in infected tissues, as well as HA-dependent itching, resulting from direct stimulation of sensory (pain) nerve endings [3,4,5]. HA also acts on endothelial cells, causing their activation and migration, which is essential for angiogenesis [6,7,8]. Much attention has recently been paid to HA in research on the physiology and pathophysiology of the gastrointestinal tract human gut microbiota due to the ability of the human gut microbiota to produce HA [9].

A breakthrough in the understanding of the role of HA in the body was the confirmation of its function as a central neurotransmitter for the brain and spinal cord [10,11,12]. In the central nervous system (CNS), HA-producing neurons are present in the tuberomammillary nucleus (TMN) of the posterior third of the hypothalamus, from which histaminergic neurons exhibit protrusions (axons) to innervate almost all CNS regions [13,14]. Histaminergic innervation plays an important role during brain development and then in adult life, participating in homeostasis and adult neurogenesis [12,13,14,15]. Therefore, there is a rationale for including disturbances in the activity of the histaminergic system in the pathomechanism of neurological diseases. This review presents the current state of knowledge concerning the involvement of HA and the histaminergic system within the CNS in the development of neurodevelopmental and neurodegenerative disorders.

## 2. Histamine

### 2.1. Histamine Biosynthesis and Metabolism

As a biogenic amine, HA is synthesized in one step from the amino acid histidine under the influence of the enzyme L-histidine decarboxylase (HDC, EC 4.1.1.22), which is the source of histaminergic signaling, via four histamine receptors (HR1–HR4).

Once formed, HA is either stored in a bound form, e.g., with heparin, primarily in the cytoplasmic secretory granules of MCs, with the possibility of rapid release upon triggering with a variety of stimuli, or undergoes immediate enzymatic degradation. HA metabolism occurs mainly through two pathways: oxidation under the influence of diamine oxidase (DAO, E.C. 1.4.3.6), leading to imidazole acetic acid (IAA), and methylation at its imidazole Nτ atom under the action of histamine N-methyltransferase (HMT, E.C. 2.1.1.8), producing tele-methylhistamine (Nτ-MH) [16]. The resulting Nτ-MH is further metabolized by monoamine oxidase B (MAO-B, EC 1.4.3.4), producing tele-methylimidazole acetic acid (t-MIAA). IAA may exist in the form of a riboside or ribotide conjugate (Figure 1).

Importantly, HA is a substrate for DAO but not for MAO because the latter enzyme converts methylhistamine (Nτ-MH). Oxidative deamination catalyzed by MAO-B is a typical transformation for all monoaminergic neurotransmitters and neuromodulators of the CNS. HA is an evolutionarily conserved signaling molecule in the CNS of vertebrates, and virtually all CNS HA is methylated; therefore, DAO concentrations in nervous tissue are very low [16,17].

### 2.2. HA Receptors

HA exerts its pleiotropic biological effects through interactions with four metabotropic HA receptor types designated H1R–H4R [18]. All HA receptors known thus far belong to the rhodopsin-like family of G protein-coupled receptors (GPCRs), also known as seven-(pass)-transmembrane (7TM) domain receptors [18,19]. This means that HA receptors interact with nearby heterotrimeric G proteins located within the plasma membrane in response to the conformational changes induced by ligand binding. HA receptor activation by ligands therefore promotes the exchange of guanosine diphosphate (GDP) for guanosine triphosphate (GTP) on the Gα subunit of the heterotrimer, resulting in its dissociation from Gβγ [20]. The classification of G proteins includes four families on the basis of the type of their α subunit: Gαs, Gαi, Gαq/11, and Gα12/13 [21]. Various Gα subunits that form trimers with the remaining Gβ and Gγ subunits can activate different signaling pathways [22,23]. All types of HA receptors are present in the CNS, with the predominant expression of H1R, H2R, and H3R in brain tissue [24,25]. Despite the demonstration of functional H4R expression in the CNS, confirmation of the presence of this receptor at the protein level was initially controversial owing to the lack of appropriate antibodies meeting strict criteria for G-protein-coupled receptors [25,26,27,28,29].

The basic characteristics of human H1R–H4R, their tissue distribution, and their function in health and disease are presented in Table 1, while their canonical signaling pathways in the context of the main effects of action are shown in Figure 2.

**Table 1 ijms-25-09859-t001:** Molecular characteristics [30,31,32,33,34,35,36,37,38,39,40,41,42,43,44,45] and biological properties [18,19,27,46,47,48,49,50,51,52,53,54,55,56,57,58,59,60,61] of the histamine (HA) receptor subtypes (H1R–H4R) in humans.

Molecular Characteristics and Biological Properties	H1R	H2R	H3R	H4R
Chromosomal location of the gene	3p25.3	5q35.2	20q13.33	18q11.2
Receptor proteins, molecular weight (MW)	487 aa, 55.78 kDa	358–359 aa, 40.1–44.5 kDa (2 isoforms)	445 aa (full-length), 36.4–49 kDa (≥20 possible mRNA isoforms)	390 aa, 34.5–44.5 kDa (2 isoforms)
Gene structure	No introns	No introns	Three introns	Two introns
Class of receptor	Rhodopsin-like GPCRs (metabotropic), seven transmembrane domain (7TM) receptors			
Binding affinity	Low (2.5 × 10^−5^ M)	Low (7.9 × 10^−6^ M)	High (6.3 × 10^−9^ M)	High (7.9 × 10^−9^ M)
Coupling subunit of the G protein complex	Gαq/11	Gαs	Gαi	Gαi
The equilibrium dissociation constant (KD)	~10 μmol/L	~30 μmol/L	~10 nmol/L	~20–40 nmol/L
Selective agonists	Histaprodiphen	Amthamine, Dimaprit, Impromidine	Imetit, Immepip, α-Methylhistamine	Imetit *, Immepip *, 4-methylhistamine, Clobenopropit (partial agonist)
Antagonists (incl. inverse agonists)	Mepyramine, Cetirizine, Chlorpheniramine, Clemastine, Diphenhydramine, Pyrilamine, Triploidine	Cimetidine, Famotidine, Nizatidine, Ranitidine	Clobenpropit, Ciproxifan, Pitolisant, Thioperamide *	2-aminopyrimidine Thioperamide, VUF-6002, JNJ-10191584, Toreforant (JNJ-38518168), JNJ-7777120, CZC-13788, PF-2988403, A-940894, A-987306
Tissue distribution	Widespread expression in many cells and tissues, including smooth muscles (e.g., in the respiratory tract and vessels), vascular ECs cells, the heart, CNS, adrenal medulla	Widespread expression in various cells and tissues, such as gastric mucosa parietal cells, smooth muscle (e.g., airways, uterus, vessels), the heart, immune cells (e.g., B cells, T cells, DCs), CNS, skin, genitourinary system, endocrine and exocrine glands	High expression in the CNS (histaminergic neurons) and neuroendocrine tissues, including enterochromaffin-like (ECL) cells in the gastric mucosa and adrenal cortex; to a lesser extent expression in the peripheral nervous system; low expression elsewhere	High expression in bone marrow, peripheral hematopoietic cells, and immune cells, including MCs, DCs, T cells, NK cells, monocytes, eosinophils; low expression elsewhere
Transmitting intracellular signals/effector molecules (see also Figure 2.)	Mobilization of intracellular Ca^2+^ levels: Gαq/11 activates PLC, which signals via DAG and PKC or IP3 to enhance Ca^2+^ release from endoplasmic reticulum, activating eNOS to release NO; H1R can activate also PLA2, leading to the formation of AA and AADE. ↑ cGMP, cAMP accumulation (via Gβγ subunits), ↑ Ca^2+^, ↑ NF-κβ, ↑ PLA2, ↑ PLD, ↑ AA, ↑ AADE	Once AC activation is initiated by Gαs, subsequent signaling occurs via cAMP and PKA; alternatively, via Gs, the H2R can activate PLC inducing DAG and IP3 pathways; Gαs promotes Ca^2+^, Na^+^, and Cl^–^ chanel opening, changing the permeability of the cell membrane ↑ Ca^2+^, ↑ c-fos, ↑ cAMP, ↑ PLC	Activated Gαi inhibits AC and modulates MAPK activity via interaction with β-arrestin 2, the recruitment of which does not depend on the formation of an active G protein complex; alternatively, activation of Gαi can inhibit K⁺ channels, Ca^2+^ channels, the Na⁺/H⁺ transporter via arachidonic acid metabolites; ↓ cAMP, ↑ Ca^2+^, ↑ MAPK, ↑ MAPK phosphorylation	Recruitment of Gαi activates PLC and inhibits membrane-bound AC activity. As a result, Ca^2+^ is mobilized from intracellular stores, while cytosolic cAMP is diminished; independently of G protein complex activation, β-arrestin 2 is recruited to the agonist-bound H4R, initiating MAPK cascade activation with subsequent receptor desensitization and internalization; ↓ cAMP, ↑ Ca^2+^, ↑ MAPK, ↑ MAPK phosphorylation
Actions mediated	- Allergic disease: ↑ APCs capacity, degranulation of MCs and basophils (release of HA and other mediators), ↓ humoral immunity and ↑ Th1 priming, ↑ Th1, ↑ IFN-γ production, ↑ cellular adhesion molecule expression, and ↑ chemotaxis of eosinophils and neutrophils, ↑ vascular permeability, hypotension; - Smooth muscle contraction (airways, intestine, uterus) resulting in bronchoconstriction, vasodilation and uterine contraction, respectively; - Nociception (↑ pruritus, ↑ burning, ↑ pain); - Thermoregulation; - Regulation of the sleep–wake cycle, food intake, locomotion, emotions (aggressive behavior), memory and learning; - Negative dromotropic and negative atrial inotropic cardiac stimulation; - Pro-angiogenic activity through H1R-PKC-VEGF-mediated pathway in EC	- Immune system: Suppression of immune cells (development of tolerance), including stimulation of suppressor T cells (Treg), ↓ neutrophil and basophil chemotaxis, ↓ neutrophil and basophil activation, ↓proliferation of lymphocytes, ↓ activity of NK cells; suppression of Th2 cells and cytokines; induction of IL-10 and suppression of IL-12 by DCs; ↑ humoral immunity; indirect role in allergic response, autoimmunity, malignancy, and graft rejection; - Changes in H2R expression accompany the process of cell differentiation - ↑ Gastric acid secretion (↑ risk of gastric ulcer); - Inflammatory response: ↑ vascular permeability, vasodilation (↑ risk of hypotension), flushing, headache; - Positive atrial chronotropic (↑ risk of tachycardia) and positive ventricular inotropic cardiac stimulation; - Bronchodilatation and ↑ mucus production (airways); - Uterus: relaxation	- Neurotransmission: presynaptic autoreceptor function—inhibition of HA release (↓ histaminergic neurons activity), as well as presynaptic heteroreceptor activity in non-histaminergic neurons—inhibition of the release of DA, 5-HT, NA (↓ sympathetic tone), ACh, GABA, glutamate; - Sympathoinhibitory action, in vivo, leads to reduced vasoconstriction, thus may promote a vasodilatory effect; - Involvement in the pathophysiology of neuroinflammation through local neuron-MCs loops; proinflammatory effect through ↑ APCs activity; - Modulation of executive functions, involvement in cognitive impairment	- Immunomodulation: involvement in DC activation and T cell differentiation; induction of proinflammatory AP-1 in Th2 cells and monocyte-derived DCs with simultaneous reduced production of the Th1-associated cytokine IL-12 and chemokine CXCL10 (IP10) in monocyte-derived DCs; modulation of eosinophil migration and selective recruitment of MCs leading to amplification of HA-mediated immune responses and subsequent chronic inflammation

↑: increase, ↓: decrease, 5-HT: serotonin, aa: amino acids, AA: arachidonic acid, AADE: arachidonic acid-derived eicosanoids, Ach: acetylcholine, AP-1: activator protein 1, APCs: antigen-presenting cells, cAMP: cyclic 3′,5′ adenosine monophosphate, c-fos: protooncogene (human homolog of the retroviral oncogene v-fos), cGMP: cyclic 3′,5′ guanosine monophosphate, CNS: central nervous system, DA: dopamine, DAG: diacylglycerol, DCs: dendritic cells, ECs: endothelial cells, eNOS: endothelial nitric oxide synthase, GABA: γ-amino butyric acid, GPCRs: G protein-coupled receptors, IFN-γ: interferon gamma, IL-10: interleukin 10, IL-12: interleukin 12, IP3: inositol trisphosphate, MAPK: mitogen-activated protein kinase, MCs: mast cells, NA: norepinephrine, NF-κβ: nuclear factor kappa-light-chain-enhancer of activated B cells, NK cells: natural killer cells, NO: nitric oxide, PKC: protein kinase C, PLA2: phospholipase A2, PLC: phospholipase C, PLD: phospholipase D, Th1 and Th2: type 1 T helper and type 2 T helper cells, respectively, VEGF: vascular endothelial growth factor. * Clobenpropit, Ciproxifan, Pitolisant, and Thioperamide are antagonists/reverse agonists of H3 receptors and therefore affect not only HA synthesis and release but also the release of other neurotransmitters, such as glutamate, ACh, DA, 5-HT, or GABA.

**Figure 2 ijms-25-09859-f002:**
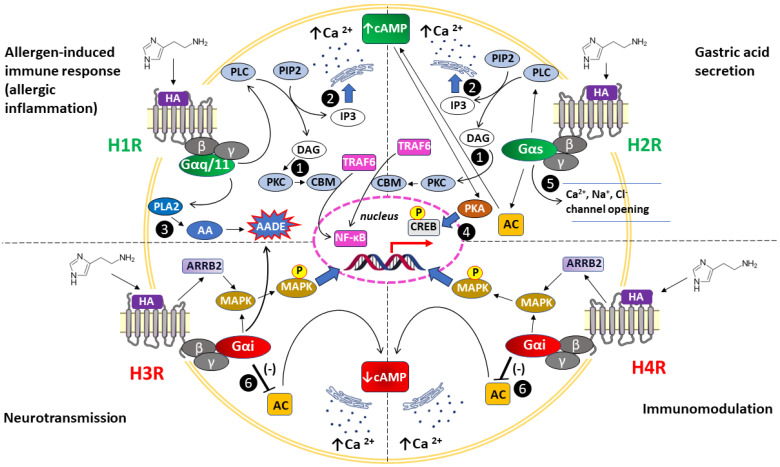
Canonical signaling pathways related to histamine H1R–H4R receptors underlying the most typical manifestations of their activity, such as allergic inflammation, gastric acid secretion, neurotransmission, and immunomodulation, respectively [19,31,33,37,39,40,43]. Due to the polymorphism of the H1R–H4R receptor genes, various alternative pathways are described, which are omitted from the figure to maintain its readability. Activation of signaling pathways for all histamine receptors is accompanied by an increase (↑) in Ca^2+^ concentration in the cytoplasm, while cAMP concentration increases (↑) after H1R and H2R stimulation and decreases (↓) after H3R and H4R stimulation. **H1R:** Gαq/11 activates PLC, which signals via ❶ DAG and PKC with the formation of CBM and—after recruitment of TRAF—downstream activation of NF-κβ transcriptional activity or ❷ IP3 to enhance Ca^2+^ release from the endoplasmic reticulum, activating eNOS to release NO; ❸ H1R can also activate PLA2, leading to the formation of AA and AADE (prostaglandins, thromboxanes, leukotrienes, and lipoxins). **H2R:** ❹ AC activation by Gαs leads to subsequent signaling via cAMP and PKA with phosphorylation of CREB in the nucleus; alternatively, the PLC is activated, which triggers signaling via DAG and IP3 (❶); ❺ Gαs promotes Ca^2+^, Na^+^, and Cl channel opening, changing the permeability of the cell membrane. **H3R and H4R:** ❻ Gαi inhibits AC and modulates MAPK activity via interaction with ARRB2, the recruitment of which does not depend on the formation of an active G protein complex; **H3R:** Alternatively, activation of Gαi can inhibit K^+^ channels, Ca^2+^ channels, and the Na^+^/H^+^ transporter via AADE (❸). AA: arachidonic acid, AADE: arachidonic acid-derived eicosanoids, ARRB2: arrestin beta-2, also known as beta-arrestin-2, cAMP: cyclic 3′,5′ adenosine monophosphate, AC: adenylyl cyclase, CBM: CBM signalosome complex = CARD11 (the caspase recruitment domain family member 11)—BCL10 (B cell CLL/lymphoma 10)—MALT1 (mucosa associated lymphoid tissue lymphoma translocation protein 1) paracaspase, CREB: cAMP-responsive element-binding protein, DAG: diacylglycerol, HA: histamine, IP3: inositol 1,4,5-trisphosphate, MAPK: mitogen-activated protein kinase, NF-κβ: nuclear factor kappa-light-chain-enhancer of activated B cells, P: phosphorylation, PIP2: phosphatidylinositol 4,5-bisphosphate, PKA: protein kinase A, PKC: protein kinase C, PLA2: phospholipase A2, PLC: phospholipase C, TRAF6: tumor necrosis factor (TNF) receptor-associated factor 6.

## 3. Cerebral HA and CNS Function

### 3.1. Importance of HA Content and Distribution in the Brain

Immunohistochemical studies have shown that the total brain HA pool consists of two compartments: the neuronal compartment, which is related to the TMN [11], and the non-neuronal compartment, the latter consisting mainly of brain-resident MCs and vascular MCs, which are closely related functionally to vascular smooth muscle cells (VSMCs) [62]. Under physiological conditions, 60–80% of cerebral HA is estimated to contain a neuronal compartment, with significant differences in HA concentrations occurring in individual anatomical structures of the brain [63]. 

Taking into account changes in the activity of histaminergic neurons and the variable number and functional state of cerebral MCs, the HA content in brain tissue shows significant topographic differences and variability in physiological and pathological conditions, including neurodevelopment [64,65,66]. This is even more understandable if we consider the role of this biogenic amine not only in modulating the immune response and neuroinflammation but also, above all, as a neurotransmitter influencing the properties of brain neurons and synapses [59,67,68,69]. Moreover, HA appears early during mammalian brain development, and this biogenic amine is one of the first neurotransmitters to appear during neurogenesis. The role of HA in the regulation of mammalian neural stem cell growth, proliferation, and differentiation has been confirmed by the fact that the highest HA concentrations are observed in the peak phase of the primary neurulation process, i.e., the formation of the neural tube from the neural plate [70,71]. Moreover, the limitation of HA synthesis in the early developmental period of the CNS is associated with underdevelopment or disruption of the corticostriatal circuitry. It has been shown that functional deficiencies of this corticostriatal signaling originating in fetal life play an important role in the pathomechanisms of Tourette’s syndrome and obsessive–compulsive personality disorder (OCPD) [72,73]. Notably, in addition to HA of neuronal origin, which is the focus of most research on brain development, non-neuronal HA may play an important role. HA and other mediators released from MCs can influence microglia-mediated neuroinflammation, which can impact brain development [62,74].

The highest HA concentrations (>3.0 pmol/mg original tissue) in the human CNS were detected in the posterior hypothalamus, which corresponds to the location of histaminergic neurons in this anatomical structure. Relatively high concentrations of HA (>1.5 pmol/mg) were also recorded in samples collected from the anterior hypothalamus, while the lowest HA levels (approximately 0.12 pmol/mg) were detected in the cerebellum and medulla oblongata. Intermediate values (in the minimum–maximum range) of HA concentrations were found in virtually all regions/anatomical structures of the brain [63]. Therefore, both the posterior hypothalamus and the anterior hypothalamus show the highest HA content in the entire CNS.

In patients with senile dementia of the Alzheimer’s disease type (SDAT), there is an increase in the content of HA in most lobes of the cerebral cortex, primarily within the functional areas of the cerebral cortex (cortical centers: sensory, motor, and association). Additionally, the increase in HA concentration compared with that in brain tissue samples collected from people whose causes of death were not related to neuropsychiatric, neurological, and/or neurodegenerative diseases was also associated with structures such as the anterior and posterior regions of the hypothalamus, putamen, caudate nucleus, nucleus accumbens, thalamus, hippocampus, pons, medulla oblongata, and cerebellum. Therefore, a relationship between excess HA and disorders of cognitive function, as well as neuroendocrine, neurovascular, and sleep–wake cycle functioning, is suggested [63]. However, there are also contrasting results, showing statistically significant reductions in the levels of HA and its precursor L-histidine in the frontal, temporal, and occipital cortices and caudate nucleus of patients with Alzheimer’s disease [74].

Elevated release and metabolism of HA has been found in the brains of chronic schizophrenia patients, which was confirmed indirectly. The increased CNS histaminergic activity was manifested by elevated levels of the HA metabolite N-tele-methylhistamine (Nτ-MH) in the cerebrospinal fluid [75].

The HA content in the CNS changes in cases of brain tissue damage. For example, increased HA levels observed in focal ischemic brain injury may play a neuroprotective role for at least 24 h after ischemia, thereby increasing the chances of recovery from neuronal damage [76,77]. Neuronal HA is also involved in pain perception and pain hypersensitivity through the sensitization of polymodal nociceptors, resulting in increased firing rates [78].

Increased release of neuronal HA occurs after stimulation of N-methyl-D-aspartic acid or N-methyl-D-aspartate (NMDA) receptors and μ-opioid receptors, as well as dopamine D2 receptors and some serotonin receptors (5-HTRs) [10].

### 3.2. Histaminergic System in the Human Brain

In all mammals studied, including humans, the only source of neuronal HA in the CNS is a loose constellation of neurons within the TMN, which are scattered around the third ventricle and the mamillary body in the ventral posterior hypothalamus [11,79]. The number of neurons within the TMN whose ability to synthesize HA from L-histidine is confirmed by the expression of histidine decarboxylase (HDC) is estimated to be in the range of 60,000–125,000 and may change in various disorders (e.g., increased in narcolepsy) [14,80,81,82]. The number of histaminergic neurons in the human brain is much greater than that in the mouse and rat brains, and they occupy a proportionally larger part of the posterior hypothalamus [83,84].

As is typical for arousal systems, HA axons of TMN neurons diffusely innervate almost all CNS regions, including the densest axonal projections, which are sent to the cerebral cortex (mainly to the frontal, parietal, and occipital lobes), hippocampus and amygdala, ventral tegmentum, nucleus accumbens and substantia nigra, midline thalamic region, striatum, cerebellum, medulla oblongata, and spinal cord [65,84,85,86]. In light of the results published by Venner et al. [87], however, the essence of the action of histaminergic TMN neurons requires further research. Doubts arose after research was conducted on the impact of GABA signaling deprivation on arousal. Importantly, the presence of the inhibitory neurotransmitter GABA was confirmed in the vast majority of histaminergic TMN neurons. However, by assessing the influence of chronic disruption of GABA synthesis and GABAergic transmission in TMN neurons on the sleep–wake cycle in male mice, the cited authors demonstrated that these neurons “are neither necessary nor sufficient for the initiation and maintenance of arousal under baseline conditions” [87]. The reason for these findings may be that only a few TMN neurons contain the vesicular GABA transporter, which is presumably necessary to release GABA. The axial spread of histaminergic tuberomammillary neurons is shown in Figure 3.

The specificity of signaling in the histaminergic system of the brain results from the fact that H3Rs are expressed within the neuron body, dendrites, and axons, as well as on the axons of other nerve cells, whereas H1Rs and H2Rs are expressed on non-neuronal target cells. H3R stimulation induces a negative feedback loop, a type of self-regulating system leading to a reduction in the synthesis and release of HA, as well as the release of other neurotransmitters, such as ACh, NA, and glutamate [65,80,88,89,90].

The functional expression of H4Rs in brain neural tissue is still insufficiently documented or nonexistent. However, H4Rs are expressed in the brain by non-neuronal cells (e.g., macrophages, MCs, and T lymphocytes), the number of which may change during various periods of life and under various pathological conditions [91,92,93].

Brain HA has been shown to be a mediator of arousal, and activation of TMN neurons promotes wakefulness, whereas TMN neurons fire little during the nonrapid eye movement (NREM) phase of sleep and are usually silent in rapid eye movement (REM) sleep [11,57,94]. The histaminergic system has also been shown to play a fundamental role in cognitive processes (e.g., learning) and the consolidation, retrieval, and expression of memory [95,96]. The results of the latest research suggest that the formation or retrieval of recognition memory requires an appropriate level of tonic activity in TMN neurons. This finding suggests that the enhancement of cerebral histaminergic signaling may promote the recovery of seemingly lost memories [97].

Electrographically documented induction of arousal is usually accompanied by disparate behaviors, including those aimed at maintaining energy homeostasis [98]. The brain’s histaminergic system modulates energy expenditure and caloric intake, also influencing thermoregulation and the circadian rhythm [23,98,99,100,101].

The fact that the H1R-H4R receptor system affects many different vital functions of the body may result not only from the variable expression of appropriate HA receptors on target cells but also from the existence of functionally specialized subpopulations of neurons within the TMN, which also show differences in terms of electrophysiology [14,79,102].

Two different functions of cerebral HA should be distinguished: neuromodulatory and classical neurotransmitters [10]. The neuromodulatory function of the histaminergic system results from the fact that most of the HA released by tuberomammillary cells does not enter the synaptic clefts but stimulates target cells via the wide diffusion pathway. A “slow” transmission mechanism is then used; after HA binds to the receptor, intracellular synthesis of the second messenger is necessary. This finding is consistent with the characteristics of histamine metabotropic receptors [103]. The “fast” effect of HA as a neurotransmitter involves its diffusion across the synaptic cleft and the activation of the H3 receptor located in the postsynaptic membrane, with a subsequent change in the functional state (opening or closing) of the controlled ion channel and a change in the permeability of the cell membrane. For example, electrical stimulation of the TMN has been shown to lead to the appearance of fast excitatory postsynaptic potentials in phasically firing neurons of the supraoptic nucleus [104]. However, owing to the significant technical difficulties associated with in vivo hemodynamic measurements, the basic neurochemical aspects of the function of HA as a neurotransmitter remain undetermined [105,106].

The brain’s histaminergic system is not isolated from other neurotransmission systems, and mutual interaction patterns may differ depending on the CNS region. Stimulation of the hetero H3 receptor inhibits the release of ACh, 5-HT, DA, and NA, whereas activation of H1R and H2R may stimulate some systems related to these neurotransmitters. The interaction of locally elevated concentrations of some of these neurotransmitters (e.g., 5-HT and DA) with their own receptors may, in turn, contribute to increased HA release [10,11,107].

Given the neuroanatomical specificity of the histaminergic system of the brain, its physiological role is related to its ability to increase neuronal excitability in virtually the entire CNS [108].

## 4. Histaminergic Dysfunction in the Developing and Adult CNS

Disturbances in HA signaling and subsequent abnormalities related to other neurotransmitters may occur both during the developmental stage of the CNS and in adulthood [109,110,111,112]. In the first case, neurodevelopmental defects usually occur, and in the second case, clinical symptoms of neurodegenerative diseases appear. However, neurodegenerative changes may be initiated at an early stage of CNS development, which is why developmental motifs are observed during the progressive and selective loss of neuronal function in adulthood [113]. In other words, some neurodegenerative diseases may be caused, or at least predefined, by disorders that arise during neurodevelopment and probably sensitize nerve cells to become susceptible to degeneration later in life [114]. Interestingly, the pathogenetic molecular mechanisms (including some key causative genes) underlying neurodevelopmental and neurodegenerative disorders overlap. These findings indicate that mitochondrial dysfunction, defects in ribonuclease protein processing, disorders of protein aggregation, synaptic plasticity, neuronal cell morphology, and the proliferative capacity of stem cells are, to varying degrees, similar in neurodegenerative and neurodevelopmental diseases [113,115].

Notably, neurogenesis not only occurs during normal prenatal and early postnatal development but also takes place in restricted brain regions (the olfactory bulb, the hypothalamus, and the hippocampal dentate gyrus) in adult mammals, including humans [116,117,118]. The histaminergic system of the brain may therefore be involved in the process, which is distinct from prenatal neurogenesis, in which neurons are generated from neural stem cells in the adult.

### 4.1. The Role of HA and Neuroinflammation during the Development of the CNS

HA, as a modulator of the properties of neurons and synapses in the brain involved in neurogenesis and gliogenesis, plays an important role at every stage of CNS development [66]. For example, in the human substantia nigra, synapses appear from the 12th week of pregnancy and mature in the 16th week [119,120]. On the basis of studies of the rodent brain, HA has been established as one of the earliest neuromodulators of embryonic development [121]. Temporary neuronal sources of HA in the early stages of CNS development, before the generation of histaminergic neurons within the TMN, have been found in various locations, including the rhombencephalon, mesencephalon, and some regions of the diencephalon [122]. This transient embryonic histaminergic system includes, among others, serotonergic neurons from the raphe nucleus that coexpress HA, thalamic neurons, and ependymal cells lining the ventricles. A characteristic feature of this system is an extensive network of histaminergic fibers in various areas of the developing brain [123]. Additionally, the HA pool is supplemented by MCs, which occur in two locations, namely the pia and the brain parenchyma [124]. Importantly, the expression of H1R, H2R, and H3R in neural stem cells (NSCs) was confirmed in vitro [125]. Consistently, H1R activation is accompanied by the promotion of NSC differentiation toward neurons, whereas H2R activation enhances the proliferation of cortical neuroepithelial cells in the presence of basic fibroblast growth factor (bFGF) [123,126].

The relationship between HA and neuroinflammation is extremely complex, especially in the case of neurogenesis [66]. First, depending on the cytokine profile, HA can locally mediate a pro- or anti-inflammatory response. Different HA concentrations may be accompanied by shifts in the cytokine composition, with a predominance of proinflammatory (e.g., interleukin 1 beta (IL-1β), IL-33, and tumor necrosis factor alpha (TNFα)) or anti-inflammatory (e.g., transforming growth factor beta (TGFβ)) cytokines [62,127,128,129,130,131] components. Further research is necessary to determine whether these different cytokine profiles are directly caused by changes in HA concentration or result from the action of other factors related to HA signaling. The source of HA may define the composition of cytokines in its environment and thus determine the neuroprotective or neurotoxic properties of this monoamine. Neuronal HA is stored in the cell body and, owing to the action of the vesicular monoamine transporter (VMAT2), in vesicles within the axonal varicosities [65,132,133]. Therefore, neuronal HA operates in a different environment than HA released after MC degranulation [134,135,136,137]. Moreover, as mentioned earlier, changes in the expression of histamine receptors and their functional state have activating (H1R and H2R) or inhibitory (presynaptic H3R) effects on both the histaminergic system and other systems via different neurotransmitters [138,139]. Therefore, in certain developmental stages of the CNS, neuroinflammation may occur with HA deficiency and, in other stages, with HA excess. Interestingly, while a properly functioning blood–brain barrier (BBB) prevents the passage of somatic (blood) HA into the adult CNS, during development, the BBB is permeable to HA [123].

The local concentration of HA may be influenced by its reuptake by both astrocytes and neurons. Both types of cells have HA transport systems related to organic cation transporter 3 (OCT3) and plasma membrane monoamine transporter (PMAT) [140,141]. In addition to their important role in regulating extraneuronal HA clearance, astrocytes are also responsible for key phenomena related to neurogenesis, such as the survival/migration of neurons, the direction of their axon growth, participation in synapse formation/maturation/elimination (pruning) and neural circuit development [142].

The spectrum of disorders and diseases arising during CNS development in which a significant role in its pathomechanism is played by neuroinflammation with abnormal histaminergic activity is constantly being supplemented [59,66,118]. These disorders can be classified into two large groups:-Neurodevelopmental disorders, including intellectual disability (intellectual development disorder), communication disorders, autism spectrum disorders (ASD), attention-deficit/hyperactivity disorder (ADHD)—the most prevalent neurodevelopmental disorder worldwide—neurodevelopmental motor disorders (e.g., Tourette’s syndrome), specific learning disorders, and schizophrenia [143,144];-Neuropsychiatric (neurodegenerative) disorders, including Parkinson’s disease (PD), Alzheimer’s disease (AD), Huntington’s disease (HD), depression, and narcolepsy [57,145,146].

The above disorders adversely affect the functioning of the affected people, impairing to varying degrees one or more spheres of functioning (e.g., personal, social, academic, and occupational), depending on the type of deficits caused. These disorders are commonly comorbid conditions, so a person affected by one of these disorders will usually meet the criteria for another disorder within the above groups [147,148]. Neurodevelopmental disorders generally appear in early childhood and can persist into adulthood. The clinical symptoms of neurodevelopmental disorders are largely the result of the formation of abnormal key neuronal circuits at the early stages of development of the nervous system, the effects of which last throughout life [149]. However, a detailed elucidation of how the dysregulation of HA itself impairs the formation/function of these neuronal circuits requires further investigation [11].

The current state of knowledge concerning the involvement of histamine signaling and neuroinflammation in the etiopathogenesis of neurodevelopmental disorders is presented in Table 2. A similar summary of neuropsychiatric diseases, which most often manifest themselves clinically after the completion of CNS development, is presented in the next subsection about neurodegenerative diseases (see Table 3).

### 4.2. The Role of HA and Neuroinflammation in Neuropsychiatric (Neurodegenerative) Disorders

An inflammatory background accompanies almost all neurodegenerative diseases in the CNS, although the specificity of this inflammation may differ significantly [170,171,172]. Moreover, many neurodegenerative diseases do not occur in isolated form but co-occur (overlap) with other disorders, including neurodevelopmental disorders [173,174]. This is due to, among other factors, the heterogeneity of neurodegenerative diseases, genotype differences, the influence of epigenetic factors, changes related to brain aging, and the co-occurrence of common diseases [175,176]. The co-occurrence of various diseases within the CNS, including the overlap of neurodegenerative disorders, may explain the fact that the degree of neurodegenerative changes indicating neuronal network disintegration in brain tissue does not directly translate into the parameters of patients’ mental assessment [177,178,179].

Neuropsychiatric disorders are easily associated with the improper functioning of histaminergic transmission pathways in the brain because HA, either directly or through other neurotransmitter systems, plays an important role in the maintenance of wakefulness, appetite regulation, mood, cognition (including learning and memory), and arousal [57]. The intensity of the inflammatory response may be crucial because hematopoietic and immune cells that flow into the CNS, such as mast cells (MCs), basophils, eosinophils, antigen-presenting cells (APCs), and CD4(+) T cells, express H4Rs [59,180]. Interestingly, under conditions of increased H4R expression in a specific location in the CNS, HA may either intensify neuroinflammation, contributing to a vicious cycle, or have anti-inflammatory effects [66,181]. Under these conditions, the direction of HA action is determined by the presence of a pro- or anti-inflammatory (damaging or neuroprotective) cytokine profile in the immediate environment.

Despite the diverse, often-not-fully understood etiology of neurodegenerative diseases, such as Parkinson’s disease (PD), Alzheimer’s disease (AD), Huntington’s disease (HD), depression, and narcolepsy, their common denominator is the presence of neuroinflammation [171,182,183]. Aberrant activation of apoptosis (programmed cell death) pathways is a common feature in these diseases that may result in unwanted loss of neuronal cells and function in a specific region of the brain [184]. Typically, neurodegenerative diseases involve the formation and intracellular deposition of protein aggregates [e.g., α-synuclein aggregations in the form of Lewy bodies (LBs) in PD patients and other α-synucleinopathies, such as dementia with LBs, multiple system atrophy, pure autonomic failure, or rapid eye movement (REM) sleep behavior disorder] due to protein misfolding or a lack of degradation, as well as excessive formation of reactive oxygen species (ROS) and reactive nitrogen species (RNS) promoting oxidative stress, endoplasmic reticulum stress (ERS), mitochondrial dysfunction, lysosomal dysfunction, alterations in calcium homeostasis, and aberrant adult neurogenesis [185,186,187].

The activation of glial cells (oligodendrocytes, microglia, and astrocytes) during the inflammatory response plays a key role in neuroinflammation, which is also due to the involvement of astrocytes in the formation and maintenance of the integrity of the BBB [88,188,189,190]. Thus, by modulating the course of neuroinflammation and affecting astrocytes, HA may indirectly influence the properties of the BBB.

Naturally, it is important to determine whether and to what extent disorders of the histaminergic system in the CNS, and in particular neurodegenerative diseases, contribute to neuroinflammation or are rather the result of neurodegeneration. This is not obvious, although attempts to influence individual histamine receptors with agonists, antagonists, and inverse agonists provide valuable information (see Chapter 5). When the involvement of the brain’s histaminergic system in the pathomechanism of neurodegenerative diseases is analyzed, the HA content in nervous tissue and the level of expression of histamine receptors of a specific type in the CNS should be considered. The role of HA and histamine receptor signaling in selected representative neurodegenerative diseases in the context of neuroinflammation is summarized in Table 3.

**Table 3 ijms-25-09859-t003:** The role of histamine (HA) signaling and neuroinflammation in the etiopathogenesis of neuropsychiatric (neurodegenerative) disorders.

Type/Name of Disorder	HA Level in the CNS (incl. Controversies)/Possible Causes or Accompanying Circumstances	Major HA Receptor Involved *	Neuroinflammation (Regardless of HA)/Typical Clinical Symptoms in Brief	References
Parkinson’s disease (PD)	Unaltered or increased. Expression of HDC-mRNA in the TMN remains undisturbed and the enzymatic activity of HDC is preserved within normal range. Moreover, the concentration of the main metabolite of HA, t-MeHA, remains unchanged in the CSF. However, abnormally high HA concentration in the basal ganglia of the brains of PD patients has been shown postmortem in other studies.	H1R, H2R, H3R	Present/Motor symptoms include the following: tremor (particularly when at rest), slowness of movement (bradykinesia), rigidity (stiffness), posture and balance problems (frequent falls), involuntary movements (dyskinesia) and muscle spasms (dystonia), speech changes (weak, hoarse, nasal, or monotonous voice, imprecise articulation, slow or fast speech, difficulty starting to speak, problems with accentuation or rhythm); non-motor symptoms include the following: anxiety and depression, stress, apathy, compulsive or impulsive behavior, dementia, and changes in sleep.	Cheng et al., 2021 [57]; Nuutinen and Panula, 2010 [69]; Shan et al., 2013 [191]; Shan and Swaab, 2022 [192]; Whalen and Gittis, 2018 [193]; Grotemeyer et al., [194]
Alzheimer’s disease (AD)	Unaltered or decreased. Despite the significant loss of histaminergic neurons in AD, expression of HDC-mRNA in the TMN remains unaltered. The decline in H1R binding may have a role in the cognitive deficits of patients with AD. Many studies have also demonstrated the involvement of H3R, whose loss of integrity and/or increased expression may locally reduce HA concentration in the CNS.	H1R, H3R	Present/The typical and common early symptom is difficulty in remembering recent events; the symptoms of dementia develop gradually over many years and eventually become more severe; in the early stages of the disease, decline in non-memory aspects of cognition, such as finding the right word, trouble understanding visual images and spatial relationships, and impaired reasoning or judgment may be observed; in the advanced-stage Alzheimer’s, patients are diagnosed with severe impairment/loss of speaking ability, disorientation (including easily getting lost), mood swings, loss of motivation, self-neglect, and behavioral issues, including lack of response to stimuli from the environment; Alzheimer’s disease is the most common type of dementia.	Zhou et al., 2024 [59]; Nuutinen and Panula, 2010 [69]; Shan et al., 2013 [191]; Shan and Swaab, 2022 [192]; Shan et al., 2012 [195]; Abdalla et al., 2023 [196]; Naddafi and Mirshafiey, 2013 [197]
Huntington’s disease (HD)	Normal to increased. TMN volume and number of TMN neurons remain within the normal range, whereas the levels of HDC mRNA, HMT activity within the inferior frontal gyrus (IFG), H1R, and H3R mRNA levels are increased. These findings indicate a functional increase in brain histaminergic signaling in HD that is linked to aberrant striatal function.	H3R, especially a hybrid dopamine–histamine receptor (D1R–H3R, heteromer), H1R	Present/Choreiform movements (jerking or writhing movements), psychiatric problems [depression, irritability/aggression, psychosis, executive dysfunction (e.g., obsessive–compulsive behaviors, apathy)], cognitive decline, and dementia (the most common genetic dementia with autosomal dominant inheritance).	van Wamelen et al., 2011 [198]; Moreno-Delgado et al., 2020 [199]; Rapanelli 2017 [200]; Jia Q et al., 2022 [201]; Angeles-López et al., 2022 [202]
Depression (also called major depressive disorder or clinical depression)	Decreased. A decrease in the HA level, especially related to the activation of H3R. Melanin-concentrating hormone (MCH) neurons express H3R through which HA directly inhibits release of MCH, an orexigenic peptide, with confirmed depressogenic effects. Moreover, other neural circuits may be influenced by lowered level of HA via H3R signaling, due to the crosstalk of HA-H3R signaling pathways with other depression-related neurotransmitters, such as 5-HT, DA, and GLU.	H3R can interact with other depression-related transmitters (including 5-HT, DA, GLU, and MCH); thus, histamine may participate in the occurrence of depression through other neural circuits.	Present/A persistent feeling of sadness, tearfulness, emptiness, or hopelessness with loss of interest or pleasure in most or all normal activities (e.g., sex, hobbies, and sports); angry outbursts, irritability or frustration, even over small matters; sleep disturbances (both insomnia or excessive sleepiness); weight loss or weight gain related to lack of appetite or overeating, respectively; loss of concentration, difficulty making decisions, slowness; feeling anxious, thinking about impending death or having suicidal thoughts; somatic ailments caused by a lower pain threshold.	Hersey et al., 2022 [203]; Qian et al., 2022 [204]; Kumar et al., 2019 [205]; Alhusaini et al., 2022 [206]
Narcolepsy [also called narcolepsy spectrum disorder (NSD)]	Increased together with the increase in the number of HA neurons in narcolepsy type 1 (narcolepsy with cataplexy) or unchanged to decreased, despite the increased number of HDC-positive neurons. In cases of elevated HA concentration, an ineffective compensatory mechanism in response to hypocretin (orexin) neurons may be suspected.	H1R by activation of hypocretin (orexin) neurons	Present/A chronic sleep disorder characterized by sudden, excessive, and uncontrollable daytime sleepiness, catalepsy (a sudden loss of control of muscle tension ranging from slight weakness to total collapse), sleep paralysis, and hypnagogic (occurring immediately before falling asleep) hallucinations.	Cheng et al., 2021 [57]; Valko et al., 2013 [81]; John et al., 2013 [82]; Dauvilliers et al., 2012 [207]; Valizadeh P et al., 2024 [208]; Shan et al., 2015 [209]; Barateau et al., 2024 [210]

* In the case of neuroinflammation, the involvement of H4R, which is expressed by increased amounts of hematopoietic and immune cells (i.e., mast cells, basophils, eosinophils, dendritic cells, and CD4(+) T cells) flowing into the CNS, should be taken into account.

## 5. Brain HA Signaling as a Therapeutic Target in Neurodevelopmental and Neurodegenerative Disorders

The complex level of interaction of histaminergic neurons with many other neurotransmitter systems makes the possibility of influencing histamine receptors within the CNS a potential therapeutic target [85,88,211]. This also applies to congenital and acquired diseases of the nervous system, the etiopathogenesis of which is still unclear, and disorders whose essence is only indirectly related to HA [57,145,212]. In such cases, treatment may only address the symptoms, not taking into account the causes of the disease (not specific), but not without effectiveness resulting from the interconnection of HA and other signaling pathways during the regulation of neuroendocrine, cardiovascular, and mental functions (e.g., arousal), including brain blood flow (vascular dynamics), BBB integrity, sleep and wakefulness, memory, cognition, learning ability, and eating and drinking behaviors [15,64,66,192,213]. Importantly, the expression of all four histamine receptors (H1R–H4R) has been demonstrated in brain tissue, but the functional expression of H4Rs has not been sufficiently documented and is rather a result of the presence of incoming immune and hematopoietic cells (e.g., MCs, T cells, and macrophages) in the CNS [25,214]. Therefore, in neurodevelopmental and neurodegenerative diseases, which are inevitably accompanied by neuroinflammation of varying severity, the role of H4Rs may be particularly important [59,215]. Simply reducing neuroinflammation may reduce brain H4R expression in these cases and thus disrupt the vicious cycle in which H4Rs modulate eosinophil chemotaxis and the selective recruitment of MCs, resulting in the amplification of HA-mediated immune responses and ultimately leading to chronic inflammation [88,216].

Modulation of the effects of HA on other brain neurotransmitter systems (e.g., DA, ACh, 5-HT, NE, GABA, glutamate, and orexin) can also be achieved by affecting H1Rs and H2Rs located on neuronal somata and dendrites [11,217,218,219]. However, the fundamental problem with the therapeutic use of H1R and H2R agonists and antagonists is that their expression is not limited to the CNS, thus causing systemic side effects [29]. This problem practically does not occur with H3Rs, whose expression is restricted to the CNS. Moreover, H3R functions not only as an autoreceptor but also as an inhibitory heteroreceptor expressed within the axon varicosities, including glutamatergic and aminergic neurons [220]. This is central to the role of H3Rs in the CNS. As presynaptic autoreceptors, H3Rs inhibit the production and release of HA from TMN neurons, and, acting as presynaptic heteroreceptors, they may also inhibit the release of other neurotransmitters (mentioned above). Interestingly, the vast majority of H3Rs are located in the postsynaptic space and on cells outside the neuronal histaminergic system. It is much more difficult to predict or interpret the effects of excitation/inhibition of postsynaptic H3Rs on the levels of HA and HA-related neurotransmitters, especially with respect to specific clinical aspects [106,221,222]. Therefore, despite the expression of all four known histamine receptors in the CNS, in recent years, only H3Rs have become the target of numerous therapeutic trials of potential novel drugs, including those developed for neurodevelopmental and neurodegenerative diseases [223,224,225].

This particular situation, when, on the basis of current knowledge about the participation of the histaminergic system and individual histamine receptors in the etiopathogenesis of brain diseases, H3R is almost exclusively selected as a direct therapeutic target and secondarily affects other histamine receptors important in a given disease, is reflected in Figure 4.

A detailed presentation of the research results on the use of numerous H3R-active substances, primarily antagonists/inverse agonists, in the context of their use in neurodevelopmental disorders and neurodegenerative diseases is beyond the scope of this review. This topic has been extensively discussed and summarized in recent years, as in the review papers of Thomas et al. [88], Ghamari et al. [226], and Szczepańska et al. [227]. Overall, the results appear to be divergent and inconclusive, ranging from promising to unsupportive of the effectiveness of a given treatment, reflecting the complexity and heterogeneity of the nervous system disorders in which these experimental therapies are used [59,228,229]. However, these studies are undoubtedly worth continuing while clarifying the indications for a specific method of treatment for a given neuropsychiatric disorder.

## 6. Concluding Remarks

The pleiotropic effects of HA are mediated by its binding to four types of membrane receptors, H1R–H4R, which are expressed in different cell types, including those occurring in the CNS. Apart from the developmental period and pathological conditions leading to disruption of the integrity of the BBB and the influx of immune and hematopoietic cells, the primary source of HA in brain tissue is histaminergic neurons located within the TMN, which send fibers to all major brain regions. The existence of the histaminergic network makes HA a conserved modulator in mammalian brains that is critically involved in many physiological functions through mutual interactions and the regulation of the activity of other neurotransmitters in the CNS, such as 5-HT, DA, NE, ACh, GABA, and glutamate. The direct or indirect involvement of histaminergic neurons in the regulation of basic homeostatic functions of the body, as well as arousal, circadian, and feeding rhythms (appetite), behavioral tasks and higher nervous activities such as cognition, inference, prediction, memory, association, consideration, speech production and understanding, and spatial orientation, has been proven.

H3Rs play a special role in modulating the activity of the central histaminergic system and other neurotransmitter systems associated with it, which results from their pre- and postsynaptic expression and autoreceptor and heteroreceptor functions. By influencing HA production and release, H3R signaling simultaneously affects H1R and H2R transmission and, in the case of neuroinflammation, H4R. Variable neuroinflammation, often accompanied by local changes in HA concentrations in brain tissue, is a constant symptom of neurodevelopmental and neurodegenerative diseases. Regardless of the etiopathogenesis of these nervous system disorders, which are usually not fully understood, there are constant attempts to improve the clinical condition by influencing HA receptors.

With few exceptions, H3R in recent years has almost exclusively become the target of numerous therapeutic trials for potentially novel drugs, including those developed for neurodevelopmental and neurodegenerative diseases. However, the results of many therapeutic concepts targeting the brain’s histaminergic system have proven uncertain or controversial, as the translation of the results of functional and behavioral studies conducted on animals to human conditions is fraught with uncertainty. Moreover, in clinical practice, it is extremely rare to encounter individual neurological diseases in isolated (“pure”) form without overlapping comorbidities. The previously mentioned dual role of HA as a neurotransmitter and mediator of the inflammatory reaction, the final neurodestructive or neuroprotective effect of which depends on the recruitment of a specific cytokine profile, also makes it difficult to clearly interpret and determine the effectiveness of a given therapy. The heterogeneity of both individual subpopulations of neurons in the TMN and the histamine receptors themselves within histaminergic neurons should also be taken into account.

The above difficulties may motivate further research aimed at developing new, detailed diagnostic methods for neurodevelopmental and neurodegenerative diseases, after which it will be possible to verify/clarify therapeutic goals related to the influence on the brain’s histaminergic system.

## Figures and Tables

**Figure 1 ijms-25-09859-f001:**
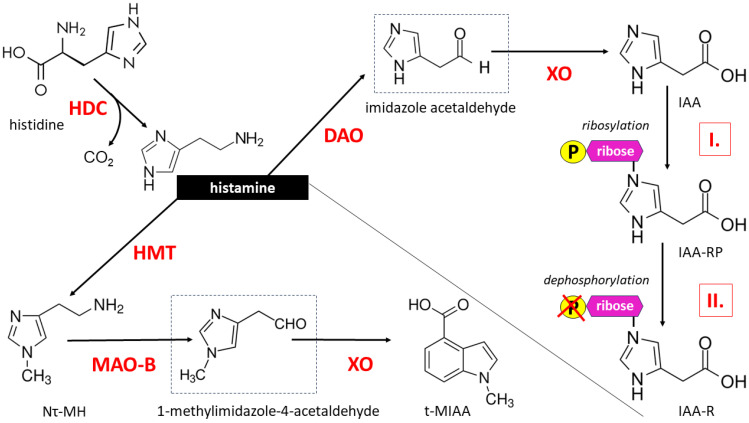
Histamine biosynthesis and metabolism. Histamine, a biogenic monoamine, is synthesized from the amino acid histidine via decarboxylation with the participation of the enzyme L-histidine decarboxylase (HDC). HA metabolism can take place through enzymatic degradation in two ways, involving two main enzymes: diamine oxidase (DAO) and N-methyltransferase (HMT). Oxidation catalyzed by DAO leads to the formation of imidazole acetic acid (IAA) and then imidazole acetic acid ribotide (IAA-RP) via imidazole acetate phosphoribosyl transferase (marked as I.)-mediated ribosylation and imidazole acetic acid riboside (IAA-R) through ribose dephosphorylation (marked with P symbol crossed out in red) catalyzed by phosphatases and 5-ecto’-nucleotidases (marked as II.). HA methylation at its imidazole Nτ atom under the influence of HMT leads to the formation of tele-methylhistamine (Nτ-MH), which is then converted to tele-methylimidazole acetic acid (t-MIAA) in a reaction catalyzed by monoamine oxidase B (MAO-B). The temporary products during the formation of IAA and t-MIAA are aldehydes (imidazole acetaldehyde and 1-methylimidazole-4-acetaldehyde, respectively) subject to xanthine oxidase (XO) action, which are marked in a dashed border. The methylation cycle plays a role in breaking down monoamine neurotransmitters including HA. Therefore, in the human central nervous system, the vast majority of histamine is methylated, while its metabolism catalyzed by DAO is negligible [16,17].

**Figure 3 ijms-25-09859-f003:**
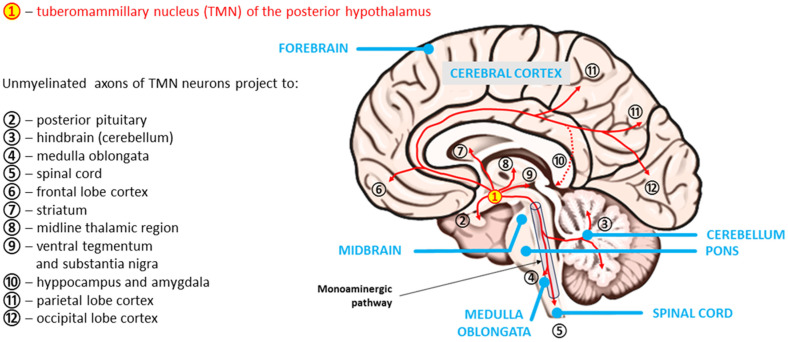
Neuroanatomy of histaminergic transmission pathways in the human brain (inspired by Haas and Panula [80], modified from Haas, Sergeeva, and Selbach [65]).

**Figure 4 ijms-25-09859-f004:**
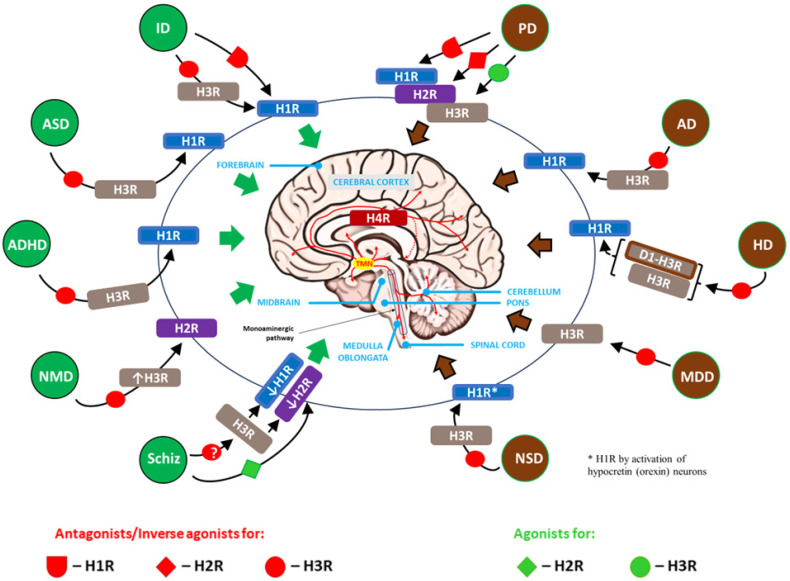
Therapeutic approach taking into account the involvement of the CNS histaminergic system in the pathomechanism of neurodevelopmental disorders (marked in green) and neurodegenerative diseases (marked in brown): histamine H3 receptor (H3R) as the main target. Other major histamine receptors have also been marked in individual disorders (see Table 2 and Table 3). All histamine H1R–H4R receptors are involved to varying degrees in the regulation of many brain processes occurring in physiological and disease states, including neurodevelopmental disorders and neurodegenerative diseases. Taking into account the neuroinflammation that accompanies these diseases, H4R was included, the functional expression of which is not sufficiently confirmed in nervous tissue but appears to be a derivative of the influx of immune and hematopoietic cells (e.g., MCs, T cells, and macrophages) into the CNS. Due to its pre- and postsynaptic expression, H3R plays a key role in the histaminergic system, regulating the release of HA and, acting on the presynaptic heteroreceptors, modulates the release of several other neurotransmitters (e.g., 5-HT, dopamine, norepinephrine, ACh, GABA, and glutamate). Furthermore, the presence of H3R confined to the CNS means that systemic symptoms that constitute side effects of H3R-acting therapy do not occur or are significantly reduced compared to direct effects on H1R and H2R. Therefore, in experimental therapeutic applications, the use of H3R antagonists/inverse agonists or agonists is seen as promising. Green arrows for neurodevelopmental disorders and black for neurodegenerative diseases indicate effects across the entire brain tissue, not a specific region. AD: Alzheimer’s disease; ADHD: attention-deficit/hyperactivity disorder; ASD: autism spectrum disorder; HD: Huntington’s disease; ID: intellectual disability; MDD: major depressive disorder; NMD: neurodevelopmental motor disorders; NSD: Narcolepsy spectrum disorder; PD: Parkinson’s disease; Schiz: schizophrenia; D1-H3R: dopamine D1-histamine H3 receptor heteromer; ↓H1R, ↓H2R: decreased expression of H1R and H2R, respectively; ↑H3R: increased expression of H3R; ?: contradictory results.

**Table 2 ijms-25-09859-t002:** The role of histamine (HA) signaling and neuroinflammation in the etiopathogenesis of neurodevelopmental disorders.

Type/Name of Disorder	HA Level in the CNS (incl. Controversies)/Possible Causes or Accompanying Circumstances	Major HA Receptor Involved *	Neuroinflammation (Regardless of HA)/Typical Clinical Symptoms in Brief	References
Intellectual disability (intellectual development disorder)	Increased due to a homozygous mutation in the histamine N-methyltransferase (HNMT) enzyme gene. HNMT is responsible for the degradation of intracellular HA.	H1R	Present/Previously called “mental retardation”; affected children start crawling, walking, or talking later than other children and have trouble with learning (the main symptom is learning slowly), communicating, thinking rationally, making judgments, and planning.	Di Marco et al., 2016 [150]; Verhoeven et al., 2020 [151]
Autism spectrum disorders (ASD)	Normal. This suggests that the response to HA is changed rather than its secretion or production in ASD. Different expression of the gene set of HNMT, H1R, H2R, and H3R may explain the different effects of HS in ASD.	H1R, H3R	Present/Symptoms of this complex condition generally appear in the first 2 years of life and include persistent challenges with social communication (e.g., decreased sharing of interests with others, difficulty assessing emotions, and lack of abstract thinking), restricted interests (inflexibility of behavior and extreme difficulty coping with change) and repetitive behavior (stereotypical movements such as hand flapping, rocking, and spinning); the limitation of normal functioning due to these disorders shows significant differences between individuals with autism.	Di Marco et al., 2016 [150]; Verhoeven et al., 2020 [151]; Griswold et al., 2012 [152]; Wright et al., 2017 [153]; Eissa N et al., 2020 [154]; Abruzzo et al., 2019 [155]
Attention-deficit/hyperactivity disorder (ADHD)	Increased due to decreased activity of diamine oxidase (DAO), the key enzyme responsible for extracellular HA degradation. Some single-nucleotide polymorphism (SNP) variants of the AOC1 gene associated with DAO deficiency or decreased DAO functionality may be involved in the pathomechanism of ADHD. The increase in brain HA may also result from SNP of the *HNMT*, encoding the main enzyme involved in intracellular metabolism of HA. It was demonstrated that the “T” allele at rs11558538 is associated with decreased HNMT activity.	H1R, H3R	Present/Executive dysfunction that can be categorized into 2 types of behavioral problems: inattentiveness (difficulty concentrating and focusing) and hyperactivity and impulsiveness	Blasco-Fontecilla 2023 [156]; Yoshikawa et al., 2019 [157]; Blasco-Fontecilla et al., 2024 [158]
Neurodevelopmental motor disorders (e.g., Tourette’s syndrome)	Decreased, due to a rare nonsense mutation, *HDC* W317X, in exon 9 of the L-histidine decarboxylase (HDC) gene. This hypomorphic mutation in the HDC gene is a rare but high-penetrance genetic cause of TS. Subsequent deficiency of HA disrupts dopamine modulation of the basal ganglia, because HA reduces dopamine levels in the stratum.	H2R and up-regulated H3R in the striatum	Present/Symptoms of motor disorders include tremors, jerks, twitches, spasms, contractions, or gait problems; these disorders can significantly limit intentional movements and cause an excess of involuntary movements, including tics (fast, repetitive muscle movements that result in sudden and difficult-to-control body jolts or sounds)	Ercan-Sencicek et al., 2010 [159]; Pittenger 2020 [160]; Baldan et al., [161]; Xu et al., 2022 [162]; Zhongling et al., 2022 [163]; Wang et al., 2023 [164]
Schizophrenia	Increased levels of N-tele-methylhistamine (Nτ-MH), a major brain metabolite of HA in the cerebrospinal fluid (CSF).	Reduced efficiency and/or decreased number of H1R binding sites, H2R (including deficit in glutamatergic neurons), H3R	Present, particularly in the dorsolateral prefrontal cortex (PFC)/A chronic mental health condition that causes a range of different psychological symptoms including reoccurring episodes of psychosis that are correlated with a general misperception of reality. The active form of schizophrenia is characterized by the following: delusions, hallucinations, disorganized speech, trouble with thinking and lack of motivation.	Ito 2004 [165]; Arrang 2007 [166]; Vallée 2022 [167]; Murphy et al., 2021 [168]; Ma et al., 2023 [169]

* In the case of neuroinflammation, the involvement of H4R, which is expressed by increased amounts of hematopoietic and immune cells (i.e., mast cells, basophils, eosinophils, dendritic cells and CD4(+) T cells) flowing into the CNS, should be taken into account.

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
