# Peer review of "Histaminergic System Activity in the Central Nervous System: The Role in Neurodevelopmental and Neurodegenerative Disorders"

_ijms, 2024, doi:10.3390/ijms25189859_

Round 1

Reviewer 1 Report

Comments and Suggestions for Authors

The present review aims to focus on the role of the histaminergic system in various central nervous system issues and neuronal degeneration. It includes fundamental information about the histaminergic system, such as histamine receptors, the distribution of histamine across different brain areas, and histamine's role in neuroinflammation, neurodevelopmental disorders, and neurodegenerative disorders. Additionally, it covers the targets of histamine in these related disorders. This review is comprehensive and engaging. To enhance its quality, the following points should be considered:

1.     In title there is no need of abbreviation, can be removed ‘CNS’.

2.     To ensure that the abstract accurately reflects the total focus of the review, it should provide a balanced overview of all the key themes discussed, not just inflammation and neuroinflammation. For example, such as the roles of histamine in neurodevelopment, neurotransmission, neuroinflammation, and neurological disorders-the abstract should briefly touch on each of all the areas. This approach gives readers a comprehensive understanding of what the review entails.

3.     The introduction also found with similar issues, as it primarily discusses the basic role of histamine in peripheral functions. However, the review is intended to focus on histamine's roles in the central nervous system, particularly its involvement in neurodevelopmental and neurodegenerative disorders. Therefore, the entire introduction should be revised to include updated and relevant information that aligns with the specific focus of this review.

4.     The statement in lines 197-198, "Under physiological conditions, 60–80% of cerebral HA is estimated to contain a neuronal compartment," needs clarification regarding whether the percentage refers to the total histamine content in the entire brain or within a specific brain region.

5.     Lines 219-225 explain the concentration of histamine in the hypothalamus. At the end of the paragraph, add a concluding statement about HA levels to reinforce the mentioned information.

6.     Lines 275-279 explain GABAergic transmission in the aerosol system. This section needs more clarification on how it influences histaminergic functions within the arousal system.

7.     Figure 3 explains the basic anatomy of histaminergic transmission pathways in the brain. It provides general information and does not include any new details, so it can be deleted.

8.     Lines 412-424 state that different levels of histamine are correlated with various inflammatory mediators, including pro-inflammatory, anti-inflammatory cytokines. This could be further elaborated to clarify whether these effects are directly mediated by changes in histamine concentration itself or if they are influenced by other factors related to histamine signaling.

9.     Line 528-531, “The activation……of the BBB”. How this paragraph related with histamine? Missing the information.

Comments on the Quality of English Language

It needs a minor improvement in flow. 

Reviewer 2 Report

Comments and Suggestions for Authors

This is a lengthy (29 pages, 229 references), comprehensive and (as far as I can discern) complete review of the complex and pleiotropic properties of histamine and histamine receptors in the mammalian brain. The (single) author accomplishes the following:

1. Histamine has both an extensive neurotransmitter and extensive neuromodulator role in the brain.

2. There are four known histamine receptors (H1R-H4R. The H4R likely is not present in brain tissue (neurons, glia, microglia) but present on inflammatory cells that can infiltrate brain cells. All histamine receptors are GPCR's and exert multiple pleiotropic actions on other neurotransmitter systems.

3. The H3R is likely both a neuronal presynaptic autoreceptor and neuronal postsynaptic receptor. H3R -selective/specific antagonists have been studied in multiple human brain conditions with variable effects. Since many of these human brain conditions are themselves heterogenous, it is not surprising that variable effects have been observed. The author makes a convincing argument for continuing to view the H3R as a therapeutic target.

4. Histamine likely plays a significant role in neuroinflammation, which is a popular subject for its involvement in many neurodegenerative diseases. There is much yet to be learned about histamine's many roles in neuroinflammation. 

Overall, in spite of the many pleiotropic actions of histamine in the brain, mediated by H1R-H3R, I found this review to be very educational and inspirational towards acquiring further knowledge about histamine functions in the CNS. The enclosed Figures (1-4) and Tables (1-3) are complex, extensive and helpful. I detected no issues with English.
